

# Combined water table and temperature dynamics control CO₂ emission estimates from drained peatlands under rewetting and climate change scenarios

Tanja Denager[1], Jesper Riis Christiansen[2], Raphael Johannes Maria Schneider[1], Peter Langen[3], Thea Quistgaard[3], Simon Stisen[1]

[1] Department of Hydrology, Geological Survey of Denmark and Greenland, Copenhagen, Denmark
[2] Forest and Landscape Ecology, Department of Geoscience and Nature Management, Copenhagen University, Denmark
[3] Department of Environmental Science, Atmospheric Emissions & Modelling, Aarhus University, Roskilde, Denmark

*Correspondence to:* Tanja Denager (tad@geus.dk)

**Abstract:**

This study integrates process-based hydrological modeling and empirical CO₂ flux modeling at a daily temporal resolution to evaluate how peatland hydrology influence CO₂ emissions under scenarios of rewetting and climate change.

Following the calibration of a three-dimensional transient groundwater flow model for a peat-dominated catchment, daily groundwater table dynamics were simulated to represent hydrological conditions in drained peat soils. These simulations were coupled with an empirical CO₂ flux model, developed from a comprehensive daily dataset of groundwater table depth, temperature, and soil CO₂ flux measurements. The empirical CO₂ flux model captures a clear temperature-dependent response of soil CO₂ emissions to variations in groundwater table depth.

By applying this coupled modeling framework, we quantified CO₂ emissions at daily timescales. The results demonstrate that incorporating both temperature sensitivity and high-resolution temporal variability in water level significantly influences projections of CO₂ fluxes. Especially the co-occurrence of elevated air temperature and low groundwater table significantly influence CO₂ emissions under scenarios of rewetting and climate change. These insights highlight the importance of including changing climate conditions in future peatland management strategies for emission inventories.

The study illustrates the value of combining detailed hydrological simulations with emission models. It also emphasizes the need for detailed monitoring of greenhouse gas emissions across multiple sites and the development of robust empirical models that can be generalized and spatially upscaled.



## Introduction

Drained peatlands are widely accepted as being net greenhouse gas (GHG) sources and rewetting of
peatlands is considered an effective means of overall net GHG emission reduction (Leifeld et al., 2019).
The depth of the groundwater table below the surface i.e. the water table depth (WTD) largely
controls the annual emissions of carbon dioxide ($CO_2$) and methane ($CH_4$) from organic soils, where
deeper WTD results in $CO_2$ emissions and a shallow WTD increases $CH_4$ emissions (Evans et al., 2021).
Despite triggering $CH_4$ emissions, rewetting of organic soils will still lead to a net long-term reduction
of GHG emissions (Günther et al., 2020). However, current estimates of GHG emissions from drained
and rewetted peatlands are still quite uncertain due to a lack of long-term monitoring and simplified
modeling approaches.
Commonly adopted methodologies for estimating contribution of organic soils in national GHG
inventories (Arents et al., 2018; Evans et al., 2021; Koch et al., 2023; Tiemeyer et al., 2020) are based
on empirical response functions between long-term annual mean WTD estimates from data-driven
machine learning (ML) models (Bechtold et al., 2014; Koch et al., 2023) and observed net ecosystem
GHG budgets (Tiemeyer et al. 2020). Those methodologies allow regional upscaling and integration
into national emission estimates.
However, significant variability in the observed net ecosystem carbon balance (NECB) used to derive
the empirical relationship can be attributed to site-specific factors, including intra-annual (seasonal)
WTD and temperature dynamics (Tiemeyer et al., 2020) caused by fluctuating climate. The current
GHG inventory methods are not suited to account for extremes such as drought and flooding that have
a profound, but temporally limited (days, weeks or months) impact on WTD. Especially the frequency
and severity of droughts can have major impacts on the $CO_2$ emissions as WTD increases together with
temperature (Olefeldt et al., 2017). Therefore, temperature changes also directly impact GHG
emissions, as soil $CO_2$ and $CH_4$ production are temperature sensitive. Currently, the impact of short-
term compound events (e.g., simultaneous warm and dry conditions (Zscheischler et al., 2020) on
annual $CO_2$ emissions from peat soil is little known. Such events can lead to consequences like a deep
groundwater table, highlighting the need for improved understanding of how climate variability and
long-term change (Olefeldt et al., 2017) affect future $CO_2$ emissions from both drained and rewetted
peatlands.
For Denmark, it is generally expected that, as a result of climatic changes, annual mean WTD will
decrease (water tables closer to terrain). However, this decrease in annual mean WTD is primarily
attributed to an increase in groundwater levels during the wetter winter months, while warmer future
summers are anticipated to experience minimal increase or even reduced summer groundwater levels
and more prolonged drawdowns (Henriksen et al., 2023; Seidenfaden et al., 2022).
The ML and statistical models of annual mean WTD (Bechtold et al., 2014; Koch et al., 2023) utilized in
current national GHG inventories (Gyldenkærne et al., 2025; Koch et al., 2023; Nielsen et al., 2025b;
Tiemeyer et al., 2020) effectively reflect the spatial variability at the national scale, but most current
ML WTD models are temporally invariant and do not account for neither inter-annual (between years)
variability, nor seasonality or intra-annual (seasonal) variability in WTD or temperature. To establish
WTD-$CO_2$ relations at intra-annual time scales, capable of capturing the impact of short-lived extreme
events such as droughts and inundations, WTD time series at these finer temporal resolutions are
required. For this, process-based transient 3D hydrological models capable of integrating unsaturated-
saturated flow models to predict spatial and temporal variability of WTD are highly useful. Combined
with the WTD-$CO_2$ relation we claim these model outputs can be used to calculate the $CO_2$ emissions
on daily, seasonal, and inter-annual timescales.





Such hydrological models provide the potential for improving our estimation of peatland hydrology
and thereby the spatio-temporal WTD variability. Improved representation of temporal variability of
WTD are needed for refining the current and future GHG estimates that cannot be derived using the
simple application of IPCC default emission factors (IPCC, 2014). Process-based hydrological models
offer the opportunity to assess the effect of different management strategies and environmental
conditions, such as rewetting and climate change.
Process-based hydrological models are increasingly being applied to study dynamics of peatland
hydrology (Mozafari et al., 2023). For example through Land Surface Models (LSM) (Bechtold et al.,
2019; Largeron et al., 2018; Shi et al., 2015; Yuan et al., 2021) utilized to analyze the soil–plant–
atmosphere exchange processes of water, energy and carbon. However, most LSM's rely on a
simplified conceptual representation of hydrologic processes and are characterized by coarse spatial
scales.
Of the studies applying fully integrated unsaturated-saturated flow models for peatland hydrology,
some focus on site or field scale models (Friedrich et al., 2023; Haahti et al., 2015; Java et al., 2021;
Stenberg et al., 2018) while others apply the models at catchment scale (Ala-aho et al., 2017; Duranel
et al., 2021; Friedrich et al., 2023; Jutebring et al., 2018; Lewis et al., 2013). A catchment scale
approach with water balance closure is particularly important for climate change impact predictions,
since the boundary conditions to the peatlands will also be affected by climate change. Similarly, the
use of catchment scale models is important because impact evaluations of peatland management
scenarios, such as rewetting, can also include impacts on streamflow and groundwater levels in
neighboring areas.
The objectives of this study were to 1) estimate current and predict the future hydrology and soil $CO_2$
emissions in a Northern European drained peatland and 2) investigate the role of rewetting and
climatic extremes on annual $CO_2$ emissions. To achieve these objectives, we used a transient
physically-based hydrological 3D model to predict daily WTD for a case study area, the Tuse Stream
catchment, representing a typical degraded Danish peatland. Secondly, we developed an empirical soil
$CO_2$ flux ($fCO_2$) model based on coupled $CO_2$ flux, WTD and temperature observations for a similar
Danish peatland (Nielsen et al., 2025a), capable of making daily predictions. Combining the
mechanistic hydrological model and the empirical emission model enabled the estimation of daily soil
$CO_2$ fluxes under current conditions as well as scenarios of rewetting and future climate, while
accounting for the impact of climatic variability and extremes.

## Data and methodology

### Study area

Tuse Stream catchment is located on the island of Zealand in the eastern part of Denmark (Figure 1a).
The total area encompasses 107 $km^2$ of which 19 $km^2$ are peat soil. The areal extent of peat soil was
determined using a national map of organic soils (Adhikari et al., 2014). The largest contiguous peat
area within the catchment is a 13 $km^2$ drained fen located in a river valley (Figure 1c) in the low-lying
part of the catchment. The peat soil area is primarily used for agriculture. In small parts of the area,
the drainage has been stopped to restore the natural hydrologic regime. The measured peat layer
thickness extends from 0.4 to 3.5 meters, below which alluvial sand deposits are typically found.
Generally, the deeper geology in the area can be characterized as clay-dominated glacial till deposits.
The catchment is characterized by flat topography, with the southern part of the catchment being
hillier. The climate conditions are humid and temperate. The catchment receives about 737 mm of
precipitation per year (1990-2024) and has an annual mean temperature of 9°C (Scharling, 1999a, b).



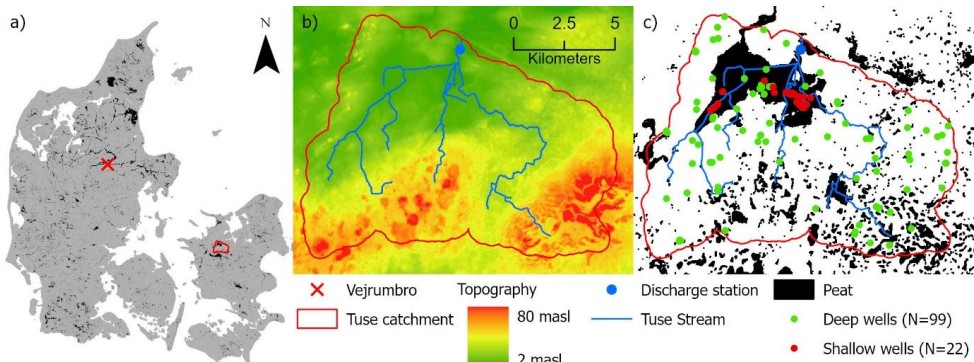

*Figure 1: a) Location of Tuse Stream catchment and the Vejrumbro site, b) topography and stream network of Tuse Stream catchment, masl: meter above sea level, c) location of organic soil and observation wells in the Tuse Stream catchment.*

Shallow WTD in the drained organic soils is monitored in 22 groundwater wells (2-3.5 meters deep) (Figure 1c). The wells are fully screened and WTD is automatic logged with pressure transducers at an hourly basis (aggregated to daily values) and verified with manual measurements. All WTD data are available in the Danish National Well Database (Jupiter, 2025). In this study, we define the water table depth (WTD) as positive when located below the terrain and negative when above the terrain. Monitoring data includes additional point measurements and timeseries of groundwater head from 99 deep wells installed in mineral soils throughout the catchment (Figure 1c). In the model setup, water extraction in 40 abstraction wells is included based on data from the Danish National Well Database in May 2020 (Henriksen et al., 2020) and implemented as yearly mean abstraction evenly distributed on the daily model timesteps. Daily discharge is monitored at the catchment outlet at Tuse Stream (Figure 1b).

## Hydrological modelling

The focus of the hydrological modelling in this study is to adequately simulate shallow groundwater levels and their dynamics for the peatland area in the Tuse Stream catchment. The fen peatland in Tuse Stream catchment is largely fed by groundwater discharge from the upstream catchment, emphasizing the need to develop a coupled groundwater surface water model at catchment scale. In addition, the objective of utilizing the model for climate change impact assessments requires a catchment scale approach with a deep groundwater component to represent changes in groundwater and surface water discharge to the peatland as well as changes in the boundary conditions. The catchment scale approach also facilitates the combined calibration and evaluation of the total water balance and peatland WTD by constraining the model with observed streamflow at the outlet as well as peatland groundwater level dynamics.

The model is set up as a transient, distributed, coupled surface-groundwater model and executed within the hydrological modeling framework MIKE SHE (DHI, 2022; Graham and Butts, 2005). MIKE SHE combines full 3D groundwater flow coupled with a gravity flow module in the unsaturated zone, 2D overland flow and 1D river flow routing in streams (DHI, 2019) (Figure S1). The simplified gravity flow module for unsaturated flow assumes a uniform vertical gradient and ignores capillary forces but provides a suitable solution for the time varying recharge to the groundwater table based on precipitation and evapotranspiration (DHI, 2022).

The model is a modified sub-model of the National Hydrological Model of Denmark (DK-model), developed at the Geological Survey of Denmark and Greenland (GEUS) (Henriksen et al., 2020; Stisen et al., 2019). The geological model is interpreted in a horizontal 100 meter grid. The numerical model is



calibrated in the same 100 meter resolution, with the saturated zone consisting of 11 computational
layers of varying thickness. The top model layer has a uniform thickness of 2 meters, which is also
applied to the peat layer areas. The bottom level of the groundwater model is defined by the
prequaternary chalk that underlies the Island of Zealand, which in the Tuse Stream catchment is
located in a depth of approximately 150-250 meters below surface.
The time-varying constant head boundary conditions at the sub-model boundary are defined from the
operational National Hydrological Model setup (Henriksen et al., 2020). The observed forcing data of
precipitation, temperature and reference evapotranspiration are provided by the Danish
Meteorological Institute (DMI) as gridded daily data in 10 km resolution for precipitation and 20 km
resolution for evapotranspiration and temperature (Scharling, 1999a, b; Stisen et al., 2011). The model
employs a maximum timestep of one day, at which the meteorological variables are fed into the
model. The model was provided with a hotstart file from an initial model run.
Spatial and temporal distributions of root depth and LAI are based on classes (Figure S2 and Table S1)
where the peat, forest, agricultural and open nature land use classes have yearly cycles of LAI and root
depth (Figure S3). Likewise, soil type is spatially distributed (Figure S2) and based on the three classes
peat, sand and clay (Table S2). In the vertical direction, the soil columns in the unsaturated zone
module are divided into 40 cells from top to bottom; 30x0.1m, 5x1m and 5x5m. Technically, the
unsaturated zone is parameterized to 33 m depth, but during simulation limited to the top of the
simulated groundwater table. We implemented uniform vertical water retention characteristics of
peat, while clay and sand water retention characteristics were defined separately for the depths 0-30
cm (horizon A), 30-70 cm (horizon B) and >70 cm (horizon C). Soil parameterization is freely adapted
from (Børgesen et al., 2009) and detailed in Table S3.
MIKE SHE allows incorporation of drainage systems, representing both artificial and natural drains. The
drainage system bypasses the slow water movement in aquifers by providing a short-cut from e.g. the
agricultural field to the nearest stream. The amount of water routed by drains from the saturated zone
to local surface water bodies is calculated using a linear reservoir model, where the difference
between groundwater head and drain level is multiplied by a drain time constant (*dt*). The drain level is
defined by a drain depth (*dd*) set relative to terrain level. Hence, drainage in any given model cell only
occurs if the simulated groundwater level exceeds the drainage level (DHI, 2022). The drain time
constant and drainage depth in each model grid cell are distributed across the model domain
according to the five land use classes (Figure S2 and Table S1).
The model parameter sensitivity analysis and subsequent calibration prioritized parameters affecting
the shallow WTD in the peat soil and the overall water balance in the catchment. A list of model
parameters can be seen in Table S3. Parameter values not included in the calibration process are
obtained from the National Hydrological Model parametrization.
Calibration method
We used the Pareto Archived Dynamically Dimensioned Search (PADDS) algorithm (Asadzadeh and
Tolson, 2013) available within the optimization toolkit Ostrich (Matott, 2019). PADDS is a multi-
objective optimizer and obtains the pareto front across multiple objective function groups, enabling
post-weighting of individual objective functions. Throughout the calibration routine, Ostrich minimized
the weighted sum of squared error (WSSE) of the objective functions. The PADDS algorithm was run
with the user settings of maximum 1000 iterations. The period 2010-2013 was used as a calibration
spin-up period and the model performance was evaluated for the 2014-2023 calibration period.
Calibration was performed against three objective function groups within the categories of WTD in
shallow groundwater wells located in drained organic soils (modified $KGE_{WTD}$), spatial correlation of the



mean WTD ($r_{spatial}$) and a combined objective function group of discharge, hydraulic head level and
seasonal amplitude in deeper wells (q_head_amp). The multi-objective optimization problem can be
formulated as:
$$\min \left( \text{q\_head\_amp}, 1 - KGE_{WTD_{modified}}, \ 1 - r_{spatial} \right) \qquad [1]$$
The metrics used as response variable in the objective functions on the three-dimensional pareto front
can be seen in Table 1.
Kling-Gupta Efficiency ($KGE_q$) was used as a conventional performance criterion for discharge. The KGE
consists of three terms: the Pearson correlation coefficient r, a term representing the measure of
variability α, and a bias term β. $KGE_q$ is included to match the overall water balance and streamflow
dynamics expressed through the discharge at the catchment outlet. $ME_{head}$ was included in the
objective function to match the general water level in the deeper aquifers across the catchment, and
$ME_{amp}$ to represent the natural seasonal variations as a mean error of yearly amplitude for hydraulic
head for deep wells with continuous time series. For a detailed description of the implementation of
$ME_{amp}$ as objective function see (Henriksen et al., 2020). KGEq, $ME_{head}$, $ME_{amp}$ were equally weighted
(according to WSSE from an initial model run) in the combination of the three metrics into one
objective function (q_head_amp).
KGE was likewise used as performance criterion for WTD, however in a modified version
($KGE_{WTD_{modified}}$). Conventionally, β in KGE is a unitless measure of the bias specified as the ratio
between the sum of simulated and observed values (β = ∑sim/∑obs). As we optimize against the WTD,
the operational sign can be both negative (water table above terrain/inundation) and positive (water
table below terrain), violating the idea of optimizing β as the ratio of sums of values with possibly
alternating operational signs. Therefore, we are using a modified $KGE_{WTD}$ where β is replaced by the
mean error (ME), see Table 1. This modification requires that the order of magnitude of the $ME_{WTD}$ is
comparable to the errors on the other terms in KGE. In our case this is ensured by the fact that the
mean observed WTD values range between approximately 0.3-0.6 m, resulting in $ME_{WTD}$ values
typically below 0.5 m. Alternatively, the $ME_{WTD}$ term could be scaled within the $KGE_{WTD}$ equation.
The calibration using the modified $KGE_{WTD}$ aims at achieving the best overall agreement between
simulated and observed WTD. Consequently, a perfect match at individual monitoring wells is not
anticipated; instead, the goal is to achieve a general fit that represents the optimal compromise across
all wells. In order to optimize the spatial variability of the mean WTD, the correlation coefficient ($r_{spatial}$)
was included as objective function.









Table 1: Objective functions metrics. KGE stands for Kling-Gupta Efficiency.

| Objective function | Observations | No. of observation wells | Metric | Abbreviation | Equation | Range | Optimum value |
|---|---|---|---|---|---|---|---|
| Modified KGE$_{WTD}$ | Daily WTD in shallow wells (in peat) | 22 | Modified KGE on WTD | $KGE_{WTD_{modified}}$ | $1 - \sqrt{(r_{WTD}-1)^2 + (\alpha_{WTD}-1)^2 + (ME_{WTD})^2}$ <br> Where, $ME_{WTD} = \frac{1}{n}\sum_{i=1}^{n} WTD_{sim_i} - WTD_{obs_i}$ | [-∞;1] | 1 |
| r$_{spatial}$ | Mean WTD over the calibration period | 22 | Spatial correlation of the mean WTD | r$_{spatial}$ | $r(\overline{WTD_{sim}}, \overline{WTD_{obs}})$ | [-1;1] | 1 |
| q_head_amp | Discharge | 1 | KGE on discharge | KGE$_q$ | $1 - \sqrt{(r_q-1)^2 + (\alpha_q-1)^2 + (\beta_q-1)^2}$ | [-∞;1] | 1 |
| | Hydraulic head in deep wells (in mineral soil) | 66 | Mean error on hydraulic heads | ME$_{head}$ | $\frac{1}{n}\sum_{i=1}^{n} head_{sim_i} - head_{obs_i}$ | [-∞;∞] | 0 |
| | | 8 | Mean error on yearly amplitude of hydraulic heads | ME$_{amp}$ | $\frac{1}{n}\sum_{i=1}^{n} A_{sim_i} - A_{obs_i}$ | [-∞;∞] | 0 |

WTD: water table depth [m], q: discharge [m/s], head: hydraulic head [m], A: amplitude [m]

A local sensitivity analysis based on initial parameter values from Table S4 was performed and values of composite scaled sensitivity (CSS) were obtained. Selection of free calibration parameters were based on the criterion that parameters were included if their CSS was larger than 0.05*CSS of the parameter with the highest CSS. The resulting 11 free parameters are indicated with grey in Table S4. Other parameters were kept at the values listed in Table S4 or tied to the calibration parameters.

## Hydrological simulations of historical and future climate

The calibrated hydrological model was run for the historical simulation period of 1990-2023 using observed climate forcing data (Scharling, 1999a, b; Stisen et al., 2011). Future hydrological projections are derived from simulations using the hydrological model forced by climate model projections, including precipitation, air temperature ($T_{air}$), and potential evapotranspiration. The resulting impacts on groundwater levels, as simulated by the hydrological model, are evaluated. We used 17 climate models (Table S5) with the Representative Concentration Pathway 8.5 (RCP8.5). The climate model outputs are generated and bias corrected by Pasten-Zapata et al. (2019), and the Global and Regional Circulation (GCM, RCM) models originate from the Euro-CORDEX project (Jacob et al., 2014).

The climate simulations cover three 30-year periods: the reference period (1991-2020), the near future (2041-2070) and the distant future (2071-2100). All 51 climate simulations (17 climate models × 3 periods) were first run using the initial potential head from the national model climate simulations (Henriksen et al., 2020). Subsequently, they were rerun using the mean potential head for the respective 30-year period as the initial potential head.

## Empirical $CO_2$ emission models

### Implementation of annual $CO_2$ emission model

Recent studies established a functional relationship between the annual net ecosystem carbon balance (NECB) for $CO_2$ and the mean annual WTD (Koch et al., 2023; Tiemeyer et al., 2020) by fitting a nonlinear Gompertz function. Like in Koch et al. (2023) and Tiemeyer et al. (2020), this study considers



NECB as only $CO_2$ fluxes, excluding methane ($CH_4$) and other carbon exports such as dissolved or
particulate organic carbon. We apply the WTD functional relationship for $CO_2$ from Koch et al. (2023),
which is fitted to Danish flux data, and refer to it as the *Annual WTD model.* The *Annual WTD model*
demonstrates a systematic relationship in which $CO_2$ flux from NECB increases with annual WTD in the
interval between 7 cm and 50 cm, above which an asymptotic level of 10 Mg $CO_2$-C ha$^{-1}$ yr$^{-1}$ is reached
(Koch et al., 2023). The *Annual WTD model* is therefore not sensitive to changes in WTD deeper than
approximately 50 cm. At WTD levels less than 7 cm, the *Annual WTD model* suggests $CO_2$ uptake;
however, this element is not included in our analysis which only models $CO_2$ emission.
Derivation and implementation of daily $CO_2$ emission model
For our empirical model to predict daily soil $CO_2$ fluxes (f$CO_2$) we assume that the WTD dependent
NECB (Tiemeyer et al. 2020, Koch et al. 2023) is driven mainly by the response of soil respiration to
WTD and $T_{air}$, as gross primary photosynthesis (GPP) and aboveground autotrophic respiration is
mostly dependent on light availability and plant phenology (Rodriguez et al., 2024). This allows scaling
to match the NECB magnitude but maintains integrity in the regulation of WTD on soil $CO_2$ fluxes.
Using a unique and comprehensive coupled dataset (Nielsen et al., 2025a) of daily mean net soil $CO_2$
fluxes, $T_{air}$ and WTD for six spatial replicate measurement points, we develop a coupled temperature
and WTD dependent empirical soil $CO_2$ flux model, hereafter referred to as the *Daily WTD-$T_{air}$ model*.
The model essentially scales the WTD-f$CO_2$ relation to $T_{air}$. The dataset Nielsen et al. (2025a) is from a
drained fen, called Vejrumbro (Figure 1), with similar characteristics (soil type, climate, land use
history) as the peat area in the Tuse Stream catchment (see methodological details in Nielsen et al.
(2025a). The soil net $CO_2$ fluxes, WTD and $T_{air}$ were measured automatically for one year (2022-2023)
(Nielsen et al., 2025a) and we used a subset of fluxes
measured for six spatial replicates 5-6 times per day, resulting in a dataset of 10950 – 13140 individual
fluxes covering 365 days (Nielsen et al., 2025a).
Implementation of $CO_2$ flux models
Spatially distributed net soil $CO_2$ fluxes are calculated at a 100-meter scale across the 13 km²
contiguous peatland area (Figure 1) with the *Annual WTD model* and the *Daily WTD-$T_{air}$ model*,
respectively, using WTD at a 100-meter scale (hectare scale) and a uniform $T_{air}$. Afterwards the
spatially distributed soil $CO_2$ fluxes are aggregated to represent the spatial mean of the 13 km²
peatland area.
First, we applied the *Annual WTD model* and the *Daily WTD-$T_{air}$ model* for the historical simulation
period of 1990-2023, using spatiotemporal distributed WTD from the calibrated hydrological model.
Afterwards, the empirical $CO_2$ models are utilized on each of the 17 climate projections for $T_{air}$ and
WTD. Daily $T_{air}$ for the Tuse Stream catchment peatland area is taken directly from the 17 bias
corrected climate projections, while daily spatial WTD is a model output from the 17 hydrological
simulations, when running the hydrological model with the forcing data (precipitation, temperature
and evapotranspiration) from the 17 climate projections. Thereby, we are able to quantify the
variability in soil $CO_2$ flux among the 17 climate projections for each of the simulation periods and
among the 30 years within each of the simulation periods.
Design and application of rewetting scenarios
For impact evaluations of peatland management scenarios, such as rewetting, on the annual $CO_2$
emissions, we define three rewetting scenarios: A, B and C. These scenarios are implemented through
controlled modifications of the simulated WTD in peatland grid cells. This method of representing
rewetting scenarios does not involve structural modifications to the hydrological model and assumes
changes in WTD without accounting for process-based feedback mechanisms within the coupled



surface–subsurface hydrological system. All rewetting scenarios are applied for the entire historical period from 1990 to 2023 and thereby representing the climatology conditions for this period and producing three 34-year time series of rewetted WTD.

The scenarios are meant to illustrate different rewetting impacts on WTD, representing wetter winters (A), uniform shift in WTD (B) and wetter summers (C), but all with the same long-term mean WTD. In Scenario A, the daily groundwater table is elevated when it is above the long-term (34-year) mean water table resulting in unchanged water table levels during summer but an increase in winter. Scenario B uniformly raises the water table by a constant scalar, while Scenario C applies the same scalar increase to water table while simultaneously reducing the annual amplitude by half. The modifications of the simulated WTD are implemented using the following equations:

$$WTD_{i_{rewet\,A}} = \begin{cases} WTD_i, & if\ WTD_i \geq \overline{WTD} \\ WTD_i + 2.5 \cdot (\overline{WTD} - WTD_i), & if\ WTD_i < \overline{WTD} \end{cases} \quad [2]$$

$$WTD_{i_{rewet\,B}} = WTD_i - (\overline{WTD} - \overline{WTD_{rewet\,A}}) \quad [3]$$

$$WTD_{i_{rewet\,C}} = \overline{WTD_{rewet\,B}} + 0.5 \cdot (WTD_{i_{rewet\,B}} - \overline{WTD_{rewet\,B}}) \quad [4]$$

where $WTD_{i_{rewet\,A}}$, $WTD_{i_{rewet\,B}}$ and $WTD_{i_{rewet\,C}}$ is the daily WTD in a grid cell for rewetting scenario A, B and C, respectively. $WTD_i$ is the daily WTD in a grid cell from the calibrated hydrological model. $\overline{WTD}$ is the long-term (34-year) mean WTD in a grid cell from the historical period of the calibrated hydrological model. $\overline{WTD_{rewet\,A}}$ and $\overline{WTD_{rewet\,B}}$ are long-term (34-year) mean WTD in a grid cell from the rewetting scenario A and B, respectively.

Bootstrapping means of future climate $CO_2$ emissions

We applied a bootstrap resampling approach to estimate the uncertainty in the mean values of soil $CO_2$ flux. Specifically, we resampled the means over the 17 climate models, each containing 30 annual values, with replacement. This process was repeated 10,000 times to construct bias-corrected and percentile-based 95% confidence intervals around the bootstrapped means.



## Results

### Hydrological model

### Calibration of the hydrological model

The model calibration, running 1000 model evaluations based on three objective function groups, using Ostrich ParaPADDS optimizer with 40 parallel model executions, took ~24 hours on a Xeon Et-4850 @2,20 GHz Server. The calibration resulted in 203 non-dominated solutions forming a three-dimensional pareto front. Figure 2 presents scatterplots of the three objective functions, illustrating the trade-offs between them. Especially, there is a clear trade-off between the two objective functions addressing temporal dynamics ($KGE_{WTD}$) and spatial dynamics ($r_{spatial}$), as illustrated in Figure 2a.

The number of non-dominated solutions and the trade-offs illustrate that several parameter sets can be considered and that an ensemble of parameter sets could be selected. For the purpose of further analysis and climate change impact assessments, however, we select one balanced solution from the non-dominated solutions, through a stepwise procedure. First, a pre-screening was performed with performance criteria for WTD of $KGE_{WTD}$ larger than 0.6, for discharge of $KGE_{discharge}$ larger than 0.6 and for hydraulic head in deeper wells of ±1 m, for $ME_{head}$ and $ME_{amp}$, respectively. Afterwards, the balanced parameter set was selected as the solution with the highest spatial correlation ($r_{spatial}$).

The selection procedure was designed to prioritize accurate simulation of the temporal dynamics of peatland WTD, while maintaining strong performance across additional objective functions and maximizing spatial correlation accuracy. Initial calibration efforts indicated that achieving a $KGE_{WTD}$ value greater than 0.6 was necessary to ensure an adequate alignment between the simulated and observed WTD time series.

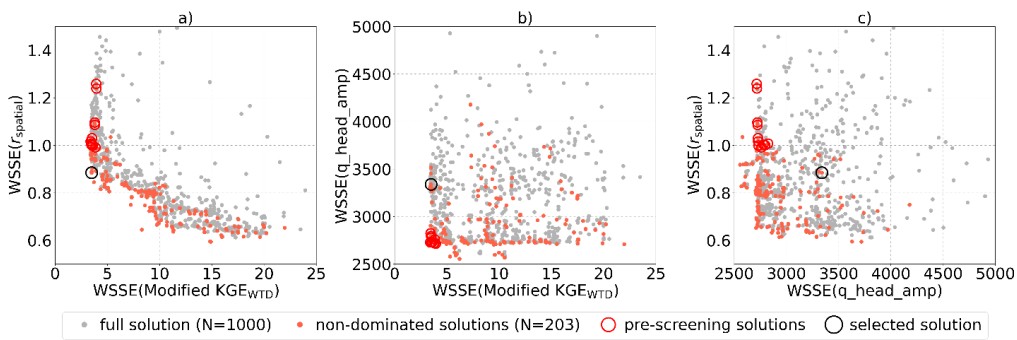

*Figure 2: Scatterplots of WSSE (weighted sum of squared errors) for the three objective function groups in the calibration. Pareto front for 1000 model evaluations.*

### Hydrological model performance

Model performance metrics for the selected solution are summarized in Table 2. The q_head_amp objective function is separated into individual contributions from the metrics $KGE_q$, $ME_{head}$ and $ME_{amp}$. Additionally, Table 2 shows the three metrics which make up the modified $KGE_{wtd}$: $r_{wtd}$, $\alpha_{wtd}$ and $ME_{wtd}$. In general, the model performs well with a $KGE_{wtd}$ in peat of 0.64, a $KGE_q$ of 0.63, a $ME_{head}$ for the deep wells of 0.75 m and a $ME_{amp}$ for the deep wells of 0.51 m for the selected solution. However, the correlation coefficient for the spatial variability ($r_{spatial}$) is poor with a value of 0.06. The model optimization achieves solid metrics on all the three components of $KGE_{wtd}$. The mean bias of WTD across all shallow peatland observation wells ($ME_{wtd}$) is only 8 cm (Table 2).



*Table 2: Hydrological model performance*

| Name of metric | | Abbreviation | Unit | Selected solution |
|---|---|---|---|---|
| Modified KGE on WTD | | $KGE_{WTD_{modified}}$ | - | 0.64 |
| | Correlation coefficient WTD | $r_{WTD}$ | - | 0.83 |
| | Measure of variance | $\alpha_{WTD}$ | - | 0.14 |
| | Mean error of WTD | $ME_{WTD}$ | m | 0.08 |
| Spatial correlation of the mean WTD | | $r_{spatial}$ | - | 0.06 |
| KGE on discharge | | $KGE_q$ | - | 0.63 |
| Mean error on the hydraulic heads | | $ME_{head}$ | m | 0.75 |
| Mean error on amplitude of the hydraulic heads | | $ME_{amp}$ | m | 0.51 |


Though the model obtains a relatively small mean error, it largely underestimates the spatial variability
in WTD. The observed mean WTD variability across the 22 monitoring wells (SD = 16.5 cm) is
considerably higher than that observed in the simulations (SD = 6.8 cm). Even though the model
performance on $KGE_{WTD}$ was generally good, it proved difficult to reproduce the spatial variation in
mean WTD.
To investigate the underestimation of spatial variability in WTD, we analyzed several spatial variables
considered relevant for explaining the observed variability in WTD: peat thickness, topography and
proximity to water bodies. However, no clear correlation was found between these spatial variables
and the mean observed WTD or model bias, as all had a correlation coefficient smaller than 0.34. See
Table S6.
Historical simulations of water table depth
The simulated WTD, generated by the calibrated hydrological model driven by historical climate for the
period 1990-2023, adequately represent both the observed seasonal patterns of WTD and their short-
term responses to precipitation events. Figure 3 shows the time series of WTD from two individual
monitoring wells as a typical example of the temporal match between observed and simulated WTD.

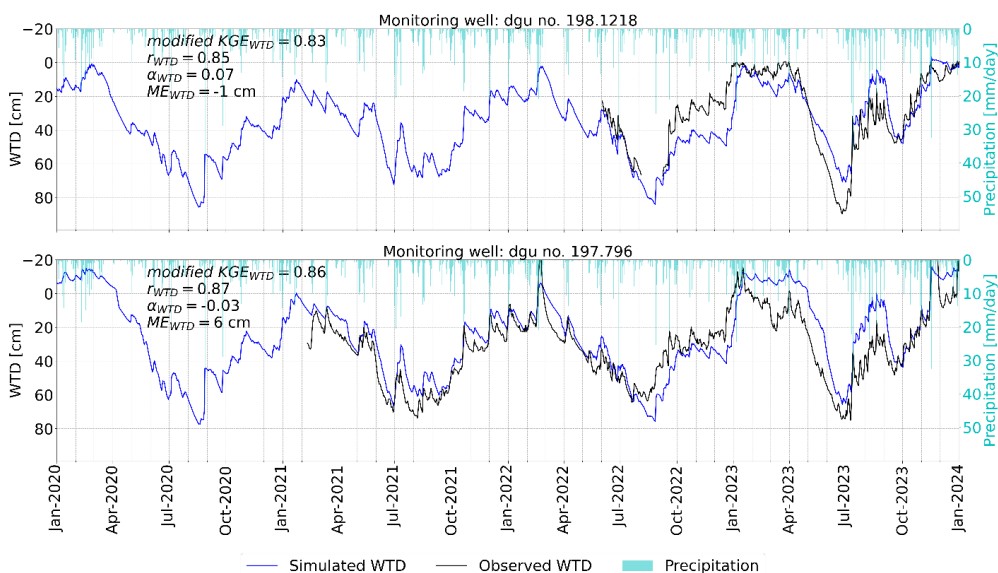


*Figure 3: Example of observed and simulated timeseries for water table depth (WTD) for monitoring wells dgu no. 198.1218*
*and dgu no. 197.796. Including metrics for these wells.*



### Meteorological climate predictions

Changes in precipitation, temperature and evapotranspiration patterns in future climate projections for Denmark generally indicate an increase in both temperature and annual precipitation. Table 3 presents the mean air temperature, mean annual precipitation and mean potential evapotranspiration derived from the 17 climate projections across the three simulation periods.

*Table 3: Mean ± SD (n=17) of annual air temperature, precipitation and potential evapotranspiration from the 17 climate models during the three simulation periods.*

|  | Unit | Reference period (1991-2020) | Near future (2041-2070) | Distant future (2071-2100) |
|---|---|---|---|---|
| Mean annual air temperature | ∘C | 8.9 ± 0.7 | 10.6 ± 0.8 | 12.1 ± 0.8 |
| Mean annual precipitation | mm yr$^{-1}$ | 780 ± 121 | 837 ± 130 | 906 ± 152 |
| Mean annual potential evapotranspiration | mm yr$^{-1}$ | 621 ± 25 | 678 ± 27 | 727 ± 27 |

### Hydrological climate predictions

Climate simulations using the hydrological model indicate a decreasing trend in mean annual WTDs (Table 4), resulting in a shallower annual mean groundwater table in future climate conditions. Both summer and winter mean WTDs are projected to be closer to the terrain, suggesting generally wetter conditions. The mean annual amplitude of WTD remains unchanged under future climate scenarios (Table 4), indicating that there is no greater seasonal drawdown of the water table during summer, although the duration of the drawdown period may be extended.

*Table 4: Statistics of WTD when using the hydrological model for climate simulations. Mean ± SD (n=17) over the 17 climate models during the three simulation periods. Summer is June, July and August, Winter is December, January and February. The amplitude is based on the monthly means of WTD to avoid outliers.*

|  | Unit | Reference period (1991-2020) | Near future (2041-2070) | Distant future (2071-2100) |
|---|---|---|---|---|
| Mean annual WTD | cm | 31 ± 1 | 27 ± 2 | 24 ± 3 |
| Mean summer WTD | cm | 47 ± 1 | 40 ± 3 | 34 ± 3 |
| Mean winter WTD | cm | 18 ± 2 | 14 ± 4 | 10 ± 3 |
| Mean annual WTD amplitude | cm | 51 ± 2 | 50 ± 4 | 52 ± 4 |

### Derivation of empirical daily soil $CO_2$ flux model

An analysis of the Vejrumbro dataset indicated a clear temperature dependency on the relation between soil $CO_2$ flux ($fCO_2$) and WTD. The Vejrumbro dataset was resampled to daily means of WTD, $T_{air}$ and soil $CO_2$ flux across the six spatial replicate measurement points omitting data from days with less than 24 flux measurements. This resulted in a dataset with 231 daily observations for each of $fCO_2$, WTD and $T_{air}$ distributed evenly over a year. Traditionally, empirical emission models for ecosystem respiration ($R_{eco}$) are fitted to soil temperature. However, due to the strong linear relationship between daily soil temperature and daily air temperature at the Vejrumbro site (r = 0.96, p-value < 0.001) (Figure S4), $T_{air}$ was used as a proxy for soil temperature when fitting the *Daily WTD-$T_{air}$ model*. This use of air temperature also facilitates upscaling and omits the need for projecting soil temperatures under climate change scenarios.

To investigate how the WTD-$fCO_2$ relation scales with temperature, we binned daily soil $CO_2$ flux into five temperature intervals: <4°C (n=39), 4-8°C (n=32), 8-12°C (n=52), 12-16°C (n=70) and >16°C (n=38) and applied a linear regression model (*y=ax*) with the intercept constrained at zero within each temperature bin. The regressions were constrained to pass through the origin, reflecting the assumption that soil $CO_2$ flux is zero when the WTD is zero. Thereby, the relationship between $fCO_2$ and WTD within each temperature bin was modeled using a linear regression of the form:



$fCO_2 = a \cdot WTD$                    [5]
where $fCO_2$ represents soil $CO_2$ flux [Mg $CO_2$-C ha$^{-1}$ day$^{-1}$], a denotes the fitted slope and WTD is water
table depth [cm], with positive values indicating depths below the terrain.
This analysis revealed an increasing slope, i.e. sensitivity of soil $CO_2$ flux to changes in WTD, with rising
temperature (Figure S5 and Figure 4a), indicating that the WTD- $fCO_2$ slope (a) can be modelled as a
linear function of temperature ($T_{air}$) (Figure 4b):
$a = b \cdot T_{air} + c$                    [6]
Combining these relationships yields a simple model of the soil $CO_2$ flux:
$fCO_2 = b \cdot T_{air} \cdot WTD + c \cdot WTD$          [7]
where $T_{air}$ [∘C] is the temperature, b [Mg $CO_2$-C ha$^{-1}$ day$^{-1}$ cm$^{-1}$ ∘C$^{-1}$] and c [Mg $CO_2$-C ha$^{-1}$ day$^{-1}$ cm$^{-1}$] are
empirical constants.

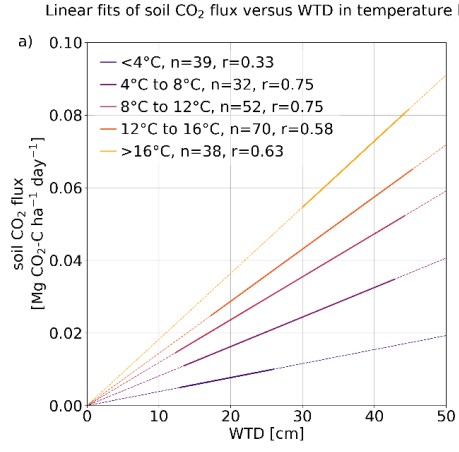
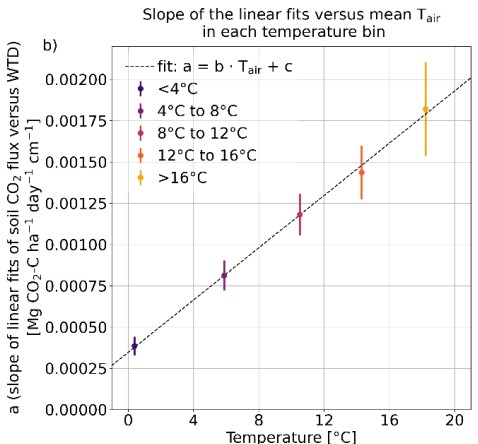


*Figure 4: Left: linear models of soil $CO_2$ flux vs. water table depth (WTD) in air temperature bins. The thicker segment of the*
*line represents the range of data used to derive the fitted model. n is the number of daily observations of soil $CO_2$ flux in each*
*temperature bin. r is Person correlation coefficient. Raw data behind the linear regressions can be seen at Figure S5. Right:*
*Slope (incl. uncertainty) (of the linear fit of soil $CO_2$ flux versus WTD) versus observed mean temperature in each temperature*
*bin.*
Having established a suitable form of the empirical soil $CO_2$ flux equation, we used nonlinear least
squares fit to estimate the b and c parameters based on the daily soil $CO_2$ flux, $T_{air}$ and WTD (without
temperature bins). This method minimizes the residual sum of squares between the observed soil $CO_2$
flux and the *Daily WTD-$T_{air}$ model*. The resulting fitted model demonstrated a significant correlation to
the observed data (r = 0.78, p-value < 0.001, RMSE = 0.021 Mg $CO_2$-C ha$^{-1}$ day$^{-1}$) (Figure S6) with daily
soil $CO_2$ flux increasing in response to rising WTD and $T_{air}$ (Figure S7). The fitted empirical constants are
as follows: b = 8.32·10$^{-5}$ Mg $CO_2$-C ha$^{-1}$ day$^{-1}$ cm$^{-1}$ ∘C$^{-1}$, c = 3.33·10$^{-4}$ Mg $CO_2$-C ha$^{-1}$ day$^{-1}$ cm$^{-1}$.
The *Daily WTD-$T_{air}$ model* predicts the highest soil $CO_2$ flux under conditions of simultaneously high $T_{air}$
and WTD, where a high WTD refers to a water table located furthest below the terrain (dry
conditions). The multiplicative *Daily WTD-$T_{air}$ model* demonstrated a moderate fit to the soil $CO_2$ flux
data, with a $R^2$ of 0.61. To assess the individual contributions of the predictor variables, we also
computed the $R^2$ between CO2 flux and $T_{air}$ and WTD separately. This was done using a constructed
dataset that included all combinations of WTD and $T_{air}$ within the model range. This resulted in $R^2$



values of 0.34 for $T_{air}$ and 0.54 for WTD (Table S7). These values reflect the explanatory power of each
variable in isolation.
Despite the significant variability in the observed net ecosystem carbon balance (NECB) used for the
*Annual WTD model* (Figure 5) it is considered to represent a robust mean as it is based on multiple
sites and years for Danish and German conditions. Compared to the *Annual WTD model* both the
measured soil $CO_2$ flux (12.9 Mg $CO_2$-C ha$^{-1}$ yr$^{-1}$ (green circle)) and the *Daily WTD-$T_{air}$* simulated soil $CO_2$
flux (13.6 Mg $CO_2$-C ha$^{-1}$ yr$^{-1}$ (not shown)) at Vejrumbro are above the corresponding fitted value of
NECB (8.7 Mg $CO_2$-C ha$^{-1}$ yr$^{-1}$ (orange circle)) based on an annual WTD of 29 cm, but still within the
range of observed NEBCs used for fitting the *Annual WTD model* (Figure 5). This may be explained by
the methodology of flux measurements at Vejrumbro that did not consider GPP ($CO_2$ uptake) and
therefore are expected to result in higher net $CO_2$ fluxes. In order to align the *Daily WTD-$T_{air}$ model* to
the level of the *Annual WTD model* where GPP is included, a scaling factor based on the above
differences ($f_{scaling}$ = 0.64) was applied to equation 7 to account for lack of GPP in the soil $CO_2$ fluxes
used for empirical model development. Applying this scaling factor, we seek to avoid the risk of
overestimating emissions when applying the *Daily WTD-$T_{air}$ model* at other locations.

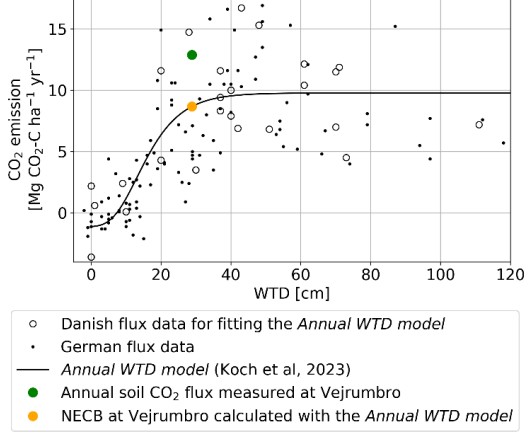


*Figure 5: The Annual WTD model together with the Danish flux data of annual NECB and WTD data underlaying the model*
*(Koch et al., 2023). German flux data are included for comparison (Tiemeyer et al., 2020). Colored circles are measured and*
*calculated soil $CO_2$ flux and NECB for the Vejrumbro dataset, so the colored circles represent the year 2022-2023.*
The Vejrumbro dataset used for fitting the *Daily WTD-$T_{air}$ model* was limited to a maximum WTD of 47
cm and maximum $T_{air}$ of 21°C (Figure S7). Outside of this range, the predictions of the *Daily WTD-$T_{air}$*
*model* exhibit increased uncertainty. It is required that the *Daily WTD-$T_{air}$ model* is sensitive to the
WTD range comparable to the annual WTD variation in the Annual WTD model. In the *Annual WTD*
*model*, the Annual NECB reaches 90% of its maximum asymptotic level at a mean annual WTD of 30 cm
(Figure 5). The mean annual WTD results from intra-annual (within year) WTD variation. The mean
annual amplitude (based on monthly means) of the observed WTD time series at the Tuse Stream
catchment study site (n=22), which is used for calibrating the hydrological model, is 65 cm. We assume
that the mean annual WTD of 30 cm originates from an annual WTD variation of ±half of that
amplitude. Therefore, we assume that the WTD sensible range of the *Daily WTD-$T_{air}$ model* is 30 + 65/2
cm = 62.5 cm. Thus, when applying the *Daily WTD-$T_{air}$ model*, daily WTD values and $T_{air}$ values were
truncated, setting WTD and $T_{air}$ to 62.5 cm and 25°C, respectively, when exceeding those thresholds.
In both the *Daily WTD-$T_{air}$ model* and *the Annual WTD model*, $CO_2$ fluxes are constrained so that the
model does not simulate negative fluxes or carbon uptake. Thus, both $CO_2$ flux models exclusively



account for the $CO_2$ emissions from the peat soil, without representing its potential role as a carbon
sink (Gyldenkærne et al., 2025).
$CO_2$ emissions from peatlands
$CO_2$ emissions throughout the historical simulation period
The long-term mean of the emission factor for the Tuse Stream catchment peat area is 8.0 ± 0.8 Mg
$CO_2$-C ha$^{-1}$ yr$^{-1}$ (mean ± SD, n=34) when using the *Annual WTD model* and 8.8 ± 1.6 Mg $CO_2$-C ha$^{-1}$ yr$^{-1}$
(mean ± SD, n=34) when using the *Daily WTD-$T_{air}$ model* (Table 5)*.*
*Table 5: Long-term mean water table depth (WTD), long-term mean annual WTD amplitude (based on monthly means of WTD*
*to avoid outliers) and long-term soil $CO_2$ flux, throughout the historical period and the three modified 34-year WTD time series*
*of rewetting scenarios. Mean ± SD is based on the 34 years of the historical period (1990-2023).*

|  | Unit | Historical period (1990-2023) | Rewetting scenario A | Rewetting scenario B | Rewetting scenario C |
|---|---|---|---|---|---|
| Mean WTD | cm | 34 ± 8 | 14 ± 18 | 14 ± 8 | 14 ± 4 |
| Mean annual WTD amplitude | cm | 51 ± 11 | 110 ± 28 | 51 ± 11 | 26 ± 5 |
| $CO_2$ emission from *Daily WTD-$T_{air}$ model* aggregated to annual | Mg $CO_2$-C ha$^{-1}$ yr$^{-1}$ | 8.8 ± 1.6 | 7.7 ± 2.0 | 5.2 ± 1.5 | 4.4 ± 0.8 |
| $CO_2$ emission from *Annual WTD model* aggregated to annual | Mg $CO_2$-C ha$^{-1}$ yr$^{-1}$ | 8.0 ± 0.8 | 4.6 ± 3.0 | 4.3 ± 2.0 | 4.4 ± 1.2 |


Figure 6 shows $T_{air}$, as wells as the spatial mean of WTD and $CO_2$ emissions across the peatland, as
simulated by the *Daily WTD-$T_{air}$ model* and the *Annual WTD model* during the historical period. The
$CO_2$ emissions calculated with the *Daily WTD-$T_{air}$ model* (red line in Figure 6c, 6d) depend on both the
observed daily temperature variability (orange line in Figure 6a) and simulated intra-annual (seasonal)
WTD variability (blue line in Figure 6b), while the $CO_2$ emission calculated with the *Annual WTD model*
(black points in Figure 6d) only depends on the inter-annual (annual means) WTD (blue points in Figure
6b) and not the temperature.
Inter-annual (between years) variation in $CO_2$ emission is substantially larger when using the *Daily*
*WTD-$T_{air}$ model* (SD = 1.6 Mg C-$CO_2$ ha$^{-1}$ yr$^{-1}$) compared to the *Annual WTD model* (SD = 0.8 Mg C-$CO_2$
ha$^{-1}$ yr$^{-1}$) (Figure 6d), as the former captures extreme events, such as periods of high temperature or
deep groundwater tables, as well as compound events involving the simultaneous occurrence of both.
In contrast, the *Annual WTD model* is insensitive to temperature and the intra-annual (within year)
timing of deep WTD. Moreover, the *Annual WTD model* imposes an upper limit of 10 Mg $CO_2$-C ha$^{-1}$
yr$^{-1}$ for annual emissions (Koch et al., 2023) (Figure 5). During the summer of 2018, a compound
extreme event occurred, characterized by both high temperatures and deep groundwater table. The
annual $CO_2$ flux for this year shows a 34% increase when estimated using the *Daily WTD-$T_{air}$ model*
compared to the *Annual WTD model*. This discrepancy arises from the *Daily WTD-$T_{air}$ model's* ability to
account for the prolonged duration of concurrent high temperatures and deep groundwater table
conditions throughout the summer (Figure 6d). Conversely, in 2010, the *Daily WTD-$T_{air}$ model*
estimates significantly lower annual $CO_2$ emissions compared to the *Annual WTD model* (Figure 6d).
This difference is due to the emission model's ability to account for the effects of prolonged periods of
low temperatures during the autumn and spring of 2010, leading to a mean annual temperature below
the long-term mean, despite summer temperatures being consistent with other years (Figure 6a).
Examples of years with extreme events primarily driven by either WTD or $T_{air}$ include 1996, which
experienced a significant summer decline in groundwater table (Figure 6b), and 1997, which was
characterized by elevated summer temperatures (Figure 6a). However, neither of these events led to




$CO_2$ emissions as high as those simulated during the compound event of both high temperatures and
deep water table in 2018 (Figure 6).

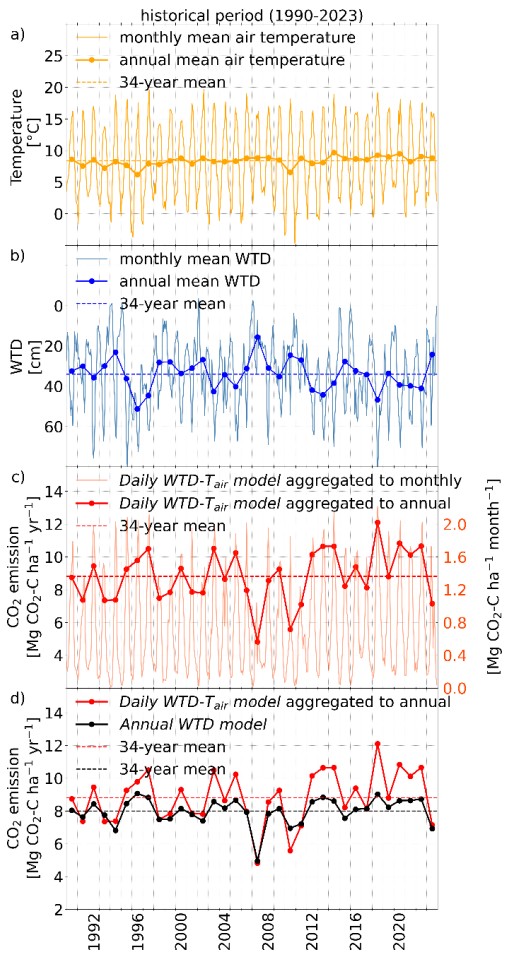


*Figure 6: Air temperature ($T_{air}$), water table depth (WTD) and soil $CO_2$ emission for the historical simulation period 1990-2023.*
$CO_2$ emissions under different rewetting scenarios
The rewetting scenarios represent an adjustment to the WTD simulated by the hydrological model
over the 34-year historical period, thereby reflecting the climatological conditions prevailing during
that time. Across all three rewetting scenarios, the long-term (34-year) mean WTD was raised by 20
cm, from 34 cm to 14 cm below the terrain, ensuring a consistent long-term annual mean WTD among
the rewetting scenarios (Table 5). Accordingly, the application of the *Annual WTD model* for estimating
$CO_2$ fluxes result in $CO_2$ emissions between 4.3 ± 1.2 Mg C-$CO_2$ ha$^{-1}$ yr$^{-1}$ (mean ± SD, n=34) and 4.6 ± 3.0
Mg C-$CO_2$ ha$^{-1}$ yr$^{-1}$ (mean ± SD, n=34) across all rewetting scenarios (Table 5). The mean annual soil $CO_2$
flux from the three rewetting scenarios, as calculated using the *Annual WTD model,* are similar but not
identical. This is because the *Annual WTD model* is applied to each of the 34 individual annual mean
WTD values rather than to a single long-term mean WTD. The SD of $CO_2$ emissions calculated using the
*Annual WTD model* in scenario C is markedly lower than in rewetting scenario A and B, reflecting the



lower inter-annual (between years) variability in mean annual WTD observed for this scenario (Table
544    5).

In contrast to the *Annual WTD model*, the *Daily WTD-$T_{air}$ model* captures the simultaneous occurrence
of low groundwater table and high $T_{air}$ during the summer months. Application of this emission model
indicates that raising the groundwater table during summer months (rewetting scenario C) yields the
greatest reduction potential in soil $CO_2$ emissions (Table 5), leading to a 50% decrease in the mean
value, from 8.8 ± 1.6 to 4.4 ± 0.8 Mg C-$CO_2$ ha$^{-1}$ yr$^{-1}$ (mean ± SD, n=34) (Table 5). In contrast,
management scenarios that primarily target increase in winter water table (rewetting scenario A)
exhibit only marginal emission reduction potential (Table 5).
A visual representation of daily soil $CO_2$ emissions in relation to mean daily temperature during the 34-
year historical period under different WTD conditions (Figure 7) reveals that high summer
temperatures are a key driver of $CO_2$ emissions. WTD observations from the Tuse catchment peatland
indicate that, during shorter periods in the warm summer months, the WTD can exceed 80 cm (Figure
3). These periods with very low summer water table contribute substantially to total $CO_2$ emissions
(Figure 7).
A rewetting scenario that mainly generates wetter winter conditions (rewetting scenario A) has very
limited $CO_2$ emission reduction. All three scenarios assume that even under rewetting, the peatland
WTD will follow a climate driven seasonality and that obtaining zero WTD in summer periods will be
difficult by classical nature-based solutions. Rewetting scenario C, which features the greatest increase
in summer WTD, achieves the largest reduction in $CO_2$ emissions (Figure 7). Permanent wet conditions
with WTD at zero would be required to obtain zero $CO_2$ emission with the developed *Daily WTD-$T_{air}$*
*model,* but under such conditions, methane emissions would also come into play and plant growth
would be severely limited.

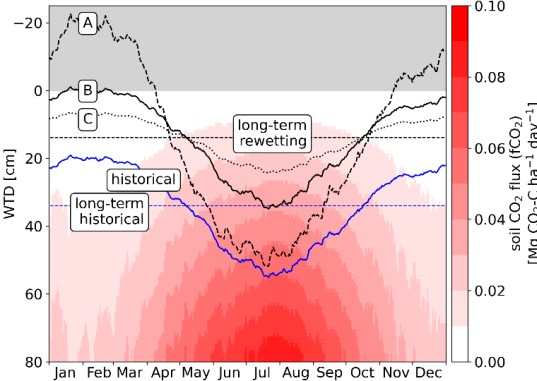


*Figure 7: Colormap: Visual representation of the annual distribution of daily surface soil $CO_2$ flux (fCO$_2$, $CO_2$ exchange with*
*atmosphere) under mean daily temperature during the historical period (1990-2023) and for different water table depths*
*(WTD). Curves: solid blue line: simulated daily mean WTD during the historical period and corresponding long-term (34-year)*
*mean WTD, black lines: daily mean WTD for each of the modified 34-year WTD time series of rewetting scenarios (A, B and C)*
*and the corresponding long-term (34-year) mean WTD.*
CO₂ emissions across future climate simulation periods
Figure 8 shows the same variables as Figure 6 but based on a representative climate model simulation
instead of the observed climate record, offering a typical example of the development of temperature,
WTD and soil $CO_2$ flux through the reference, near and distant future periods based on the RCP 8.5
pathway.





The future climate simulations show an increase in both the annual mean temperature and
groundwater levels combined with higher maximum summer temperature (Figure 8a, 8b, Table 3,
Table 4). The bootstrap mean of soil $CO_2$ flux calculated with the *Annual WTD model* over all climate
models predicts a decreasing trend in soil $CO_2$ flux under future climate conditions (Figure 9a,
horizontal dotted black line in Figure 8d), driven by an inter-annual (between years) mean WTD closer
to terrain (Table 4, Figure 8b). However, this decreasing trend is countered by the inclusion of $T_{air}$
effects when applying the *Daily WTD-$T_{air}$ model* (Figure 9b, horizontal dotted red line in Figure 8c and
8d*)*.
The wider confidence intervals in the mean annual $CO_2$ emissions for the future periods with both $CO_2$
emission model (Figure 9) indicate that the inter-annual (between years) soil $CO_2$ fluxes become more
variable in future climate. Furthermore, the confidence intervals for the individual periods are wider
for the *Daily WTD-$T_{air}$* (Figure 9b) compared to the *Annual WTD model* (Figure 9a), which is expected as
variations in $T_{air}$ and not only WTD is included as with the *Daily WTD-$T_{air}$ model*. This demonstrates that
the *Daily WTD-$T_{air}$ model* captures extreme events, including periods of high temperature or deep
groundwater table, whether these events occur simultaneously (compound event) or independently.

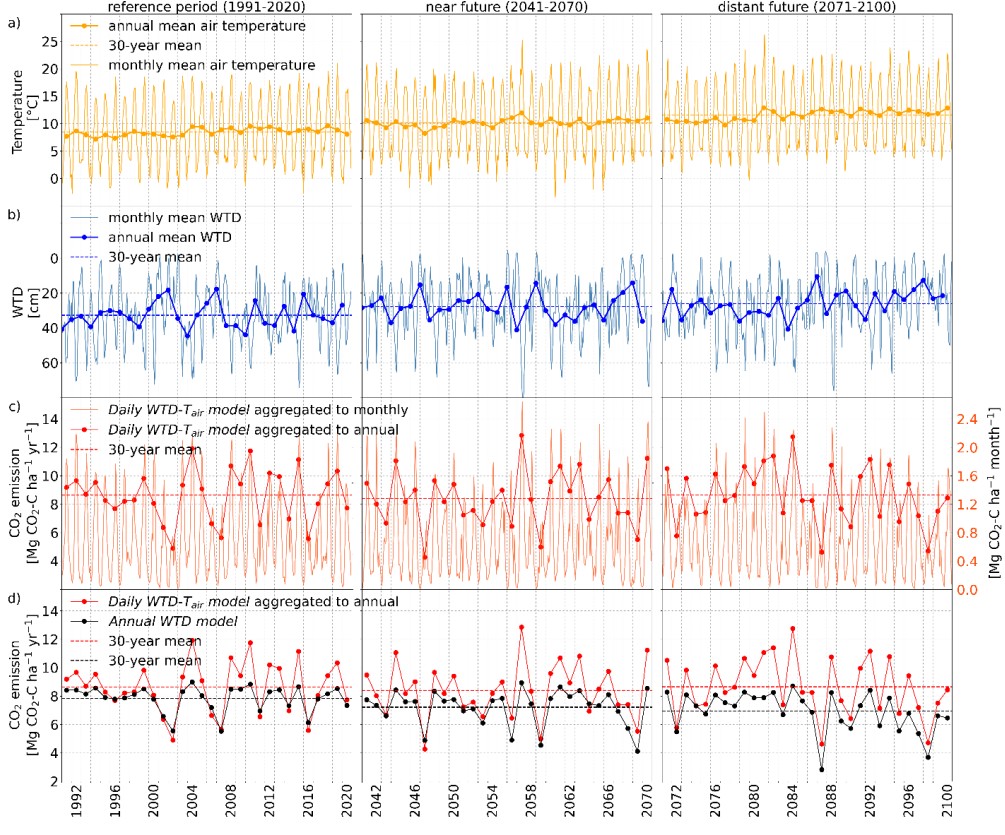


*Figure 8: Example of air temperature ($T_{air}$), water table depth (WTD) and soil $CO_2$ flux for future climate simulation with*
*climate model projection no. 5 (Table S6).*




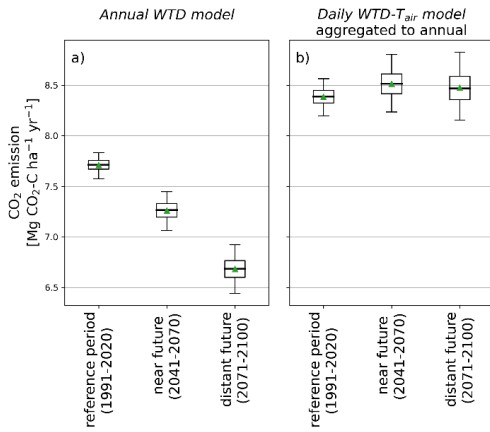

*Figure 9: Boxplot showing the distribution of bootstrap means of soil $CO_2$ emissions according to the Daily WTD-$T_{air}$ model and Annual WTD model during future climate. Green triangles and horizontal lines indicate the mean and the median of the bootstrap mean, respectively. Boxes show the $25^{th}$ and $75^{th}$ percentiles. Whiskers indicate the 95% confidence intervals. Outliers are not shown.*

The results presented in Figure 9 suggest that the impact on $CO_2$ emissions caused by future increases in $T_{air}$ and increases in water tables cancel each other out when using the *Daily WTD-$T_{air}$ model*. To investigate this further, we analyze how the combination of $T_{air}$ and WTD shift between the reference and the distant future periods, despite relatively stable total $CO_2$ emission.

We wish to identify the specific combination of $T_{air}$ and WTD that are associated with the majority of the $CO_2$ emission. Due to the non-linear response of soil $CO_2$ flux to environmental drivers in the *Daily WTD-$T_{air}$ model*, a large fraction of total emissions is generated on relatively few days. To quantify this, we calculated p50, defined as the proportion of days required to account for 50% of the total annual soil $CO_2$ flux ($fCO_2$). This was achieved by ranking the daily values of $fCO_2$, WTD, and $T_{air}$ in ascending order according to $fCO_2$. Subsequently, the ranked $fCO_2$ values were cumulatively summed to obtain their percentile distribution (Figure S8). The procedure was first applied to $fCO_2$, WTD, and Tair data from the historical simulation period, with the resulting percentile curves shown in Figure S8. Over the historical simulation period, 50% of the total $fCO_2$ ($fCO_{2, p50}$) was generated within 22% of the days (p50 = 22%), while the value of $fCO_{2, p50}$ and corresponding $WTD_{p50}$ and $T_{air, p50}$ are estimated to be $4.15·10^{-2}$ g $CO_2$-C ha$^{-1}$ day$^{-1}$, 47 cm and 13.8 °C (Table 6 and Figure S8).

Similar estimates are derived from the three timeslots from the climate models (reference, near- and distant future) using the 17 different climate models. For the future, 50% of the total $fCO_2$ is expected to occur within approximately 21 ± 1 % (mean ± SD, n=17) of the days (Table 6). The daily soil $CO_2$ flux associated to p50 ($fCO_{2, p50}$) and p50 are nearly identical across both the historical and future climate simulations periods (Table 6). As also shown in Figure 9b, the magnitude and temporal distribution of $fCO_2$ are predicted to remain unchanged in the future. While the value of $fCO_{2, p50}$ remains relatively constant around $4·10^{-2}$ Mg $CO_2$-C ha$^{-1}$ day$^{-1}$ for future climate periods, the corresponding $WTD_{p50}$ and $T_{air, p50}$ values change as a result of changing climate moving towards higher temperatures (17 °C) and shallower groundwater table (40 cm).

Figure 10 provides a graphical representation of $fCO_2$ obtained from the *Daily WTD-$T_{air}$ model*, with the colormap illustrating the daily $fCO_2$ corresponding to different combinations of $T_{air}$ and WTD. The daily $fCO_{2, p50}$ ($4.15·10^{-2}$ g $CO_2$-C ha$^{-1}$ day$^{-1}$ for the historical period (Table 6)) can be achieved through various combinations of $T_{air}$ and WTD (dark red dotted line in Figure 10). The values of $T_{air, p50}$ and



$WTD_{p50}$ corresponding to $fCO_{2,\,p50}$ for the Tuse Stream catchment peatland are plotted as a dark red
point. As expected, the $fCO_{2,\,p50}$ values for the reference periods of the 17 climate models (green
crosses at Figure 10) are closely aligned with that of the historical period. It is evident that the $fCO_{2,\,p50}$
values for the distant future climate conditions (blue crosses at Figure 10) shift along the direction
indicated by the pink arrow (along the red dotted line), reflecting a trend toward higher temperatures
and lower WTD (i.e. water levels closer to the terrain surface). This indicates that the mean daily $fCO_2$
(Table 6) and the long-term $fCO_2$ remains constant in the future (Figure 9b), as a result of a
counterbalance between impacts of rising temperatures and rising groundwater levels.
The pink arrow at Figure 10 illustrates the characteristic impact of climate change in Denmark,
reflecting the concurrent increase in air temperature and shallow groundwater levels (Schneider et al.,
2022). In contrast, other regions in Europe are experiencing declining groundwater level trends to
climate change (Wunsch et al., 2022). Consequently, $CO_2$ emissions from peatlands in these regions
are expected to shift in the direction indicated by the yellow arrow in Figure 10, towards considerably
larger emission rates.
*Table 6: p50 is the fraction of days required to reach 50% of the total soil $CO_2$ flux ($fCO_2$). $fCO_{2,\,p50}$ is the daily soil $CO_2$ flux*
*associated with p50. $WTD_{p50}$ and $T_{air,\,p50}$ are the water table depth (WTD) and air temperature ($T_{air}$) corresponding to $fCO_{2,\,p50}$,*
*respectively. Mean ± SD is based on 17 climate model simulations.*

|  |  | Historical simulation period | Climate simulation periods | | |
|---|---|---|---|---|---|
|  |  |  | Reference period | Near future | Distant future |
|  | Unit | (1990-2023) | (1991-2020) | (2041-2070) | (2071-2100) |
| p50 | % days | 22 | 21 ± 1 | 21 ± 1 | 21 ± 1 |
| $fCO_{2,\,p50}$ | Mg $CO_2$-C ha$^{-1}$ day$^{-1}$ | $4.15\cdot10^{-2}$ | $4.03\cdot10^{-2} \pm 9.89\cdot10^{-4}$ | $4.00\cdot10^{-2} \pm 3.24\cdot10^{-3}$ | $4.03\cdot10^{-2} \pm 3.65\cdot10^{-3}$ |
| $T_{air,\,p50}$ | °C | 13.8 | 14 ± 0.3 | 15 ± 0.6 | 17 ± 1.0 |
| $WTD_{p50}$ | cm | 47 | 46 ± 1 | 42 ± 3 | 40 ± 3 |


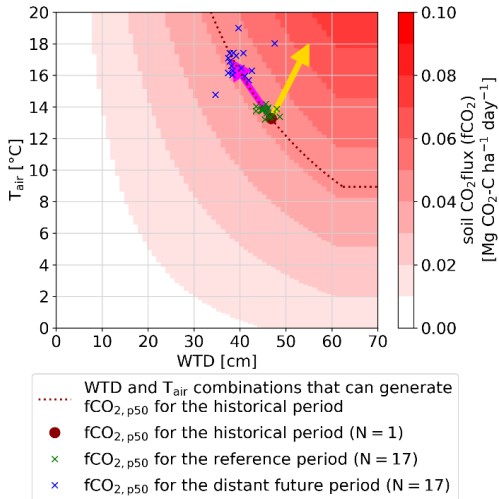


*Figure 10: Colormap: Visual representation of the Daily WTD-$T_{air}$ model output, illustrating soil $CO_2$ flux (fCO2) as function of*
*daily water table depth (WTD) and air temperature ($T_{air}$). The dark red dotted line represents combinations of $T_{air}$ and WTD*
*that corresponds $fCO_2$ at p50 ($fCO_{2,\,p50}$), where p50 is the fraction of days required to reach 50% of the total accumulated $fCO_2$*
*during the historical period. Green crosses are $fCO_{2,\,p50}$ for the reference period of the 17 climate simulations. Purple crosses*
*are $fCO_{2,\,p50}$ for the distant future period of the 17 climate simulations. The pink and yellow arrows indicate different future*
*trends in $T_{air}$ and WTD and the associated trend in $CO_2$ emissions under climate change. Specific to Denmark, the pink arrow*



*indicates increases in $T_{air}$ and decrease in WTD, other regions might experience increase in both $Ta_{ir}$ and WTD and an*
*associated large increase in $CO_2$ emissions (yellow arrow).*



## Discussion

### Peatland management under changing climate

In 2023, $CO_2$ emissions from drained organic soils in croplands and grasslands was estimated to accounted for 6.7% of Denmark's total emissions, including those from the Land Use, Land-Use Change and Forestry (LULUCF) sector (Nielsen et al., 2025b). Returning peatland organic soils to their natural hydrological state is a cost-effective GHG reduction strategy (IPCC, 2014; Kirpotin et al., 2021; Tanneberger et al., 2021; Wilson et al., 2016). Therefore, national policies (Regeringen, 2024) and the European Union's Nature Restoration Law (Regulation (EU) 2024/1991, 2024) seek to improve the management of peatlands and achieve climate neutrality targets under the urgent Green Transition agenda. To mitigate agricultural GHG emissions Danish ministerial agreements were initiated in 2024, targeting the restoration of 140,000 hectares of peatland (Regeringen, 2024). However, there is a need to strengthen the scientific evidence for mitigation measures to facilitate cost-effective policies. Quantitative predictions of fluxes such as the numbers presented in this study, supports prioritization and design of peatland rewetting strategies by estimating their $CO_2$ emission reduction potentials accounting for future climate variability impact on $CO_2$ emissions.

Integration of the process-based hydrological model of the Tuse Stream catchment with the empirically derived *Daily WTD-$T_{air}$ model* of soil $CO_2$ flux developed in this study revealed that emission simulations at daily timesteps produce greater variability in soil $CO_2$ fluxes compared to emission estimates derived from annual WTD means. This increased variability is attributed to the daily model's ability to account for short-term compound events, especially the simultaneous occurrence of elevated air temperatures and low groundwater levels.

More importantly, incorporating temperature dependence and higher temporal resolution into the $CO_2$ emissions model significantly alters the projected trends of $CO_2$ emission under both rewetting and changing climate conditions. The rewetting analyzed in this study showed how different rewetting scenarios with varying seasonal amplitudes in WTD suggest significantly different emission reduction potential even with identical annual mean WTD. The results illustrate that increasing the groundwater table during warm periods is key to obtaining $CO_2$ emission reductions, whereas rewetting strategies that mainly raise winter water table without significantly affecting the summer levels offer limited mitigation benefits. This highlights the importance of not only targeting annual reductions in WTD but particularly designing rewetting strategies to increase the summer water table and avoid critically low water levels during droughts and warm periods. Achieving such rewetted conditions may include larger forced control of WTD than what is currently being practiced for most existing rewetting schemes, where the WTD remain subject to climate seasonality impact. With such nature-based solutions it is not likely to reduce $CO_2$ emissions to the degree that current emission reduction policies target.

Also, projections of $CO_2$ emissions under different climate change scenarios were altered greatly by introducing temperature sensitivity and enhanced temporal resolution into the $CO_2$ emissions modeling framework. Here our results show that, while the projected rise in groundwater tables in isolation would lead to lower $CO_2$ emissions in future (when using the *Annual WTD model*), the *Daily WTD-$T_{air}$ model* revealed that anticipated increases in $T_{air}$ are likely to cancel out these reductions, resulting in $CO_2$ emissions on a level comparable to current levels. This is an important finding, since it suggests that increasing temperatures alone will likely increase $CO_2$ emissions, and that water level rise driven by climate change or rewetting initiatives might just counteract this trend. Rewetting measures would need to be substantially intensified to ensure climate resilience and achieve meaningful reductions in $CO_2$ emissions. Additionally, outside the specific case of Danish peatlands located in a region that is susceptible to a future wetter climate, other regions might project both increasing



temperatures and lower groundwater tables, and in such cases climate change will significantly
increase emissions without any rewetting.
Hydrological simulation of groundwater levels in peat soil with process-based models
Existing large scale $CO_2$ emission estimates, such as national inventories from organic soils
(Gyldenkærne et al., 2025; Nielsen et al., 2025b), typically combine empirical emission models and
data-driven ML approaches for estimating annual WTD (Bechtold et al., 2014; Koch et al., 2023;
Tiemeyer et al., 2020). These approaches appear robust and suited for upscaling but are limited in
their ability to represent the impact of sub-annual variability in temperature and WTD, which are
issues that become increasingly important when analyzing effects of rewetting and climate change. In
contrast to most data-driven approaches, hydrological models enable a climate-driven representation
of WTD temporal dynamics and the underlying hydrological processes. Moreover, the use of physically
based models has the distinct advantage of enabling scenario-based analyses, such as the evaluation
of alternative land use strategies and the projection of future hydrological conditions under climate
change scenarios. Utilizing hydrological models that generate high-resolution time series of WTD, it is
possible to quantify impacts of WTD dynamics, including water levels, temporal variability and
seasonal amplitudes, on changes in $CO_2$ emissions.
A unique feature of the present study is that the hydrological model of Tuse Stream catchment is
developed in the same modelling framework as the National Hydrological Model of Denmark
(Henriksen et al., 2020; Stisen et al., 2019). The National Hydrological Model is continuously updated
with new data and operates in near real-time. This integration enables a link between the lessons
learned from the Tuse Stream catchment-scale model and the National Hydrological Model of
Denmark, thereby improving the representation of peatland hydrology and contributing to the
refinement of future national GHG inventories.
As a continuation of this study, we will further investigate the spatial variability of WTD and extent
hydrological model to include additional peatland-dominated catchments. Additionally, we will utilize
the National Hydrological model to simulate WTD across all Danish peatlands.
Selection, fit and transferability of daily $CO_2$ emission model
A range of empirical models with varying levels of complexity has been developed to describe
ecosystem respiration; however, the most commonly applied formulation is the Lloyd–Taylor model
(Lloyd J., Taylor, 1994), in which temperature acts as the sole independent variable. Structural
complexity in empirical equations is increased through the integration of various other environmental
variables, for example, hydrological variables such as WTD (Rigney et al., 2018).
To evaluate alternative empirical emission models alongside our *Daily WTD-$T_{air}$ model,* we fitted three
different empirical formulations from Rigney et al. (2018) to the Vejrumbro soil $CO_2$ flux data (Table
S7). Each of the three empirical formulations incorporated both temperature and WTD as independent
variable. The model fitting resulted in $R^2$ values comparable to those obtained from fitting the *Daily
WTD-$T_{air}$ model* developed in this study (Table S7).
Studying the explanatory power of each independent variable of WTD and $T_{air}$ in isolation in the other
empirical emission models, revealed that models in which WTD and $T_{air}$ are incorporated as additive
terms, rather than as interdependent (e.g., multiplicative) terms (as in eq. 6 and 8 in Rigney et al.,
(2018)), often exhibit coefficients of determination ($R^2$) that are excessively dominated by either WTD
or $T_{air}$ (Table S7). This indicates that such model formulations may inadequately capture the joint or
synergistic effects of these variables on the dependent variable. The challenge likely stems from the
fact that both WTD and $T_{air}$ exhibit similar seasonal patterns, which may lead the regression to
primarily fit one of the additive terms containing either WTD or $T_{air}$. Empirical models that incorporate



WTD and $T_{air}$ as multiplicative terms (such as equation 7 in Rigney et al. (2018) and the *Daily WTD-$T_{air}$*
model developed in this study) demonstrate a more balanced distribution of explanatory power
between each independent variable (Table S7). Nevertheless, equation [7] in Rigney et al. (2018)
remains predominantly influenced by the $T_{air}$ component (Table S7). A more balanced distribution of
explanatory power between temperature and WTD is desirable, given that both variables are
recognized as key drivers of soil $CO_2$ flux dynamics, which is achieved better with the *Daily WTD-$T_{air}$*
than with any of the empirical models in Table S7.
In this study, we demonstrate the need for the development of emission models operating on a sub-
annual timescale. It highlights the necessity of creating scalable generalized models based on
temperature, WTD and possibly other predictors. The development of such models requires data from
a large number of sites with continuous and temporally dense measurement, in order to integrate
information in a manner similar to models based on annual WTD. We recognize that currently, models
based on annual WTD are likely the most robust for upscaling to national level and current conditions.
The simulated soil $CO_2$ flux at Vejrumbro, estimated using the *Daily WTD-$T_{air}$* model (13.6 Mg $CO_2$-C ha$^{-1}$
$^1$ yr$^{-1}$), aligns well with flux measurements from Danish and German sites (Figure 5). This agreement
suggests a comparable magnitude of emissions across geographically distinct locations of similar
characteristics, such as soil type and land use history.
We acknowledge that the *Daily WTD-$T_{air}$ model* is derived from a single dataset, and that other
emission models also provide valid fits of WTD and $T_{air}$. Furthermore, we recognize that empirical
emission models are highly dependent on the specific data to which they are fitted. Acknowledging the
limited data behind the *Daily WTD-$T_{air}$ model* utilized in this study, the goal has not been to accurately
estimate the peatland emission budget, which will be uncertain due to the reliance on a single site.
However, the objective has been to illustrate the impact and insights gained from applying emission
models at a daily timescale and how this has significant impact on the conclusions that can be made
regarding effects of rewetting and climate change. The decision to utilize the *Daily WTD-$T_{air}$ model* for
rewetting and climate modeling scenarios is motivated by the simplicity of the relationship and its
direct derivation from the Vejrumbro data, which clearly demonstrates a temperature-dependent
relationship between soil $CO_2$ flux and WTD. The limited availability of multiple high-temporal-
resolution GHG emission datasets broadly restricts the ability to generalize and upscale empirical GHG
emission models at a daily timescale. Therefore, we consider the *Daily WTD-$T_{air}$* model to be the most
reliable option currently available. Future research should validate the performance of emission
models on intra-annual (within years) data with continuous measured $CO_2$ data.
A promising methodology for future applications, as well as for integrating a Tier 3 framework,
involves coupling a process-based hydrological model with process-based emission models or an
empirically derived daily emission model, such as the one developed in this study, to enable detailed
simulations of GHG emissions that capture short-term dynamics and compound environmental effects.




## Conclusion

This study demonstrates the feasibility of simulating the temporal dynamics of the peatland water
balance and shallow groundwater table depth (WTD) using a catchment-scale distributed hydrological
model. Accurately modelling shallow WTD is critical for reliable projections of $CO_2$ emissions from
peatlands. We combined simulations of shallow WTD from the calibrated hydrological model with two
empirical $CO_2$ emission models 1) an annual WTD-$CO_2$ relationship and 2) a daily WTD-$CO_2$ model
accounting for the temperature effect on soil $CO_2$ production. This approach was used to estimate net
soil $CO_2$ emissions for the historical period (1991-2020), the near future (2041-2070) and the distant
future (2071-2100). This demonstrated that projections of soil $CO_2$ emissions are highly sensitive to the
complexity and temporal resolution of the emission model applied. Specifically, models that
incorporate both temperature and WTD dynamics at a daily timescale results in vastly different
conclusion regarding impacts of climate change and rewetting. Regarding climate change impacts, we
show that a daily temperature and WTD based emission model predict increased emissions due to
temperature changes, which can be counter balanced (in the Danish case) or amplified depending on
the future trend in WTD. Our results also demonstrate that rewetting strategies aimed at raising the
groundwater table during the warm summer period offer a $CO_2$ emission reduction potential of up to
50%, whereas approaches focused primarily on increasing winter water table levels result in only
marginal reductions. The combination of process-based hydrological model simulations and a daily-
resolution empirical $CO_2$ emission model used in this study captures the influence of short-term
compound climate events—such as simultaneous high temperatures and low WTD—which
substantially alters projected emission trends compared to simpler approaches. Such refined
approaches are essential for developing adaptive, climate-resilient peatland restoration policies and
improving national greenhouse gas inventories. The findings underscore the importance of moving
beyond static, annual WTD thresholds in peatland management by incorporating dynamic hydrological
simulations. Instead, rewetting strategies should prioritize maintaining elevated summer groundwater
table levels to buffer against drought-induced emission peaks.

## Supplement link

811 ...

## Author contributions

All authors contributed to the conception and design of the study. TD conducted the analysis and
drafted the manuscript, with input and revisions from all co-authors.

## Competing interests

The authors declare that they have no conflict of interest.

## Acknowledgements

We would like to thank Independent Research Fund Denmark for supporting the project PEAtlands and
Climate-driven variability in groundwater depth – Impacts on greenhouse gas Emissions.

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
