# Peer review of "Combined water table and temperature dynamics control CO2"

_EGUsphere, 2025_

## Author Comment (AC1)

**Reply to RC1:**

The manuscript presents a well-structured and scientifically sound study with valuable findings on the prediction of CO2 emissions from peatlands under warming climate and rewetting scenarios. The modeling approach and discussion are generally well executed. I recommend the authors consider the following minor revisions to improve clarity and consistency:

- Lines 63–66: The sentence is difficult to follow due to the use of both WTD and groundwater level, which change in opposite directions. Please consider reformulating the sentence using only WTD.
- Lines 70–71: Correct the double negative in "not ... neither ... nor."
- Table S3: Please define the variables Alpha, N, L, and Kint.
- Line 201: The term KGEWTD is not defined—please add a definition.
- Line 203: To maintain consistency with other objective functions, consider using subscripts for q\_head\_amp.

We appreciate your suggestions and will revise the sentences to enhance clarity and define the terms more precisely.

• **Lines 215–217**: It is unclear how the objective functions are combined, given their differing ranges and optimal values. Please clarify the methodology.

We understand the confusion. Below we have rewritten section "Calibration method" (previously Lines 191-245) to ensure that methodology is clearly explained:

**"Calibration method**

[revised manuscript text omitted]

"

• **Tables S3 and S4**: Both tables present water retention parameters for peat, but with differing values. Could you explain the reason for this discrepancy?

We apologize for this discrepancy. Some values originate from an earlier model calibration, and the inconsistent values will be corrected.

• Line 287: Remove the unnecessary line break.

**Certainly, we will.**

• Line 433: The temperature sensitivity of soil fCO2 is commonly reported as non-linear. While the selected model and alternatives are well addressed in the discussion, I'm curious—did you explore temperature sensitivity within WTD bins? The model appears to miss many of the observed high fCO2 values; perhaps a non-linear temperature dependence could improve performance?

I am unsure whether you meant exploring temperature sensitivity within **WTD bins** or within **temperature bins**. In the paper we explored the WTD sensitivity to  $fCO_2$  within temperature bins. Figure R1 illustrates the relationship between  $fCO_2$  and temperature within temperature bins (as defined in the article), and Figure R2 presents the relationship between  $fCO_2$  and temperature within WTD bins. I agree that the observed data clearly indicates a non-linear relationship between  $fCO_2$  and temperature.

However, the relationship between fCO2 and temperature in our developed WTD-Tair model is also non-linear. In Figure R3, I have plotted the observed fCO2, the fCO2 simulated using the WTD-Tair model, and the fCO2 simulated using the Lloyd-Taylor model (Rigney et al., 2018, Eq. 5) against temperature. The results clearly show that the WTD-Tair model follows the Lloyd-Taylor model but provides a better match with the observed fCO2, as it also accounts for WTD sensitivity ( $r_{\text{WTD-Tair model}} = 0.78$ ,  $r_{\text{Lloyd-Taylor model}} = 0.63$ ).

The very high observed values, e.g. in Figure R3, will not be possible to capture with a simple model based solely on Tair and WTD, since we also observe much lower fCO2 rates on other days with similar Tair and WTD (see Figure S5), so the model becomes a compromise that captures some variability while preserving a sound mean.

I agree that it would be valuable to investigate whether the non-linear relationship between  $fCO_2$  and temperature within bins could be incorporated into our emission model to improve its performance especially for high values of  $fCO_2$ . However, I also believe that our developed WTD-Tair model is simple (which is a strength in itself) and performs well, even though it underestimates the highest  $fCO_2$  values. Therefore, we consider the WTD-Tair model sufficient for the purpose of this study.

Figure 1: temperature sensitivity within temperature bins

Figure 2: temperature sensitivity within WTD bins

Figure 3: observed and simulated  $fCO_2$  versus observed temperature for the observed  $fCO_2$ ,  $fCO_2$  simulated with the WTD-Ta model and  $fCO_2$  simulated with the Lloyd-Taylor mode.

• Lines 479–487: While I understand the need to limit fCO2, the rationale behind the chosen thresholds (WTD 62.5 cm and Tair 25°C) is unclear. I expect hese values to strongly influence the comparison of modeled fluxes for extreme years (Lines 516–517). Please elaborate.

In the Daily WTD–Tair model, fluxes are sensitive to the WTD threshold but less affected by the temperature threshold. As you also note, we needed to constrain  $fCO_2$  in the Daily WTD–Tair model. Our rationale for the selected threshold is explained in lines 479–487, where we aimed to choose a reasonable value based on data and comparability to the sensitivity range of the annual model. Currently, we lack a more accurate approach to define the WTD threshold because daily  $fCO_2$  data for WTD values deeper than 47 cm are unavailable (line 475).

Below we have tried to update the reasoning behind the selected thresholds for WTD and Tair (previously Lines 479-487):

"The Vejrumbro dataset used for fitting the *Daily WTD-Tair model* was limited to a maximum WTD of 47 cm and maximum Tair of 21°C (Figure S7). Outside this range, the predictions of the *Daily WTD-Tair model* exhibits increased uncertainty. At the same time, it is generally understood that the upper portion of the peat layer drives the net CO2 emissions observed at the surface. Therefore, the extrapolation of WTD in the *Daily WTD-Tair model* must be constrained. The *Daily WTD-Tair model* should be sensitive within a WTD range comparable to the expected daily variation in the Annual WTD model, which also reaches an fCO2 asymptotic at deeper water tables. In the *Annual WTD model*, the Annual NECB reaches 90% of its maximum asymptotic level at a mean annual WTD of 30 cm (Figure 5). The mean annual WTD results from intra-annual (within year) WTD variation described by the annual amplitude. The mean annual amplitude (based on monthly means) is 65 cm, across the 22 observed WTD time series in the Tuse Stream catchment used for calibrating the hydrological model. We assume that a mean annual WTD of 30 cm originates from an annual WTD variation with a similar amplitude. Therefore, we assume that the WTD range of the *Daily*

*WTD-Tair model* is 30 + 65/2 cm = 62.5 cm. For the Tair range, it is assumed that the sensitivity continues until  $25^{\circ}$ C, which is a daily average value very rarely occurring, even in future climate projections. Thus, when applying the *Daily WTD-Tair model*, daily WTD values and Tair values were truncated, setting WTD and Tair to 62.5 cm and  $25^{\circ}$ C, respectively, when exceeding those thresholds."

We carried out a sensitivity analysis to evaluate the impact of the WTD threshold (Figure R4). The impact of the WTD threshold is clear, and as expected the WTD threshold has a greater influence on the extreme year than the average years. On average the emission estimate changes by app. 10% for average years and 14% for the extreme dry year as a function of WTD threshold with a reasonable range of thresholds. It should be noted that these differences would apply to all scenarios for climate and rewetting, e.g., in Figure 7 and 9 and therefore would not change the conclusions regarding impact of model selections, only the total emission estimates. The sensitivity to the Tair threshold is much smaller, mainly because daily average temperatures above 20°C very rarely occur.

Figure R4: Proportional change in mean  $CO_2$  emission from Daily WTD-Tair model aggregated to annual for different WTD threshold values. 62.5 cm is the threshold value in the preprint. Blue: modelled mean  $CO_2$  emissions across 34 years. Orange: modelled  $CO_2$  emissions for the extreme year 2018.

• Lines 488–491: The statement that "both CO2 flux models exclusively account for the CO2 emissions from the peat soil" is misleading. The Annual model includes NEBC, and thus GPP, as discussed earlier in the manuscript. GPP is also used as a reason for downscaling the daily model for fCO2. Please revise this statement for accuracy.

You are right, we appreciate your input and will adjust the sentence as recommended.

• Figure 6: Consider combining panels c and d, as they contain overlapping information.

We agree, there is overlapping information. However, the repetition of the daily model aggregated to annual values serves as reference in both plots, both to the variations in Tair and WTD and to the alternative Annual model. In order to not make panel c too "busy" by adding the annual model, we prefer to keep both panels.

**Discussion**: The section is generally well structured. However, it would benefit from additional citations, e.g., regarding the advantages of hydrological models and process-based emission models.

In the revised manuscript, we will include a more profound discussion on the use of hydrological and emission models, along with relevant citations.

---

## Author Comment (AC2)

**Reply to RC2:**

The authors have done hydrological modelling of peatland WTD and its impacts on CO₂ fluxes in current and future climates under different rewetting scenarios. Overall the study is well conducted and clearly represented. The study is highly topical considering the importance of peatland ecosystems in GHG budgets and the open questions still surrounding them. I only have a few minor comments on the manuscript.

1. Concerning the rewetting scenarios: Are there actual management practices, that can focus the rewetting seasonally? I am only aware of the more nature-based methods, such as ditch-blocking. I appreciate that discussing actual, real-life management practices is not the point of this manuscript but it would perhaps be helpful to somewhere explain how these scenarios used here relate to real life, as your findings rather strongly suggest that considering the seasonal variation in rewetting is important.

In some cases, management practices place impermeable membranes along the edges or surrounding peatlands. These artificial interventions aim to sustain higher water levels, especially during summer, than would be achieved through nature-based solutions. These artificial engineered management practices are especially relevant in the context of bogs as opposed to fens. Such practices can also ensure that areas outside the project site, such as neighboring agricultural fields, are not affected by the rewetting. However, it is an expensive solution.

In addition, the outcome of this study can serve as a reference for discussions on realistic expectations on CO2 emission reductions. Especially, in cases where nature-based solutions are planned, and where a return to more natural hydrology will still encompass temporal variability and climate induced droughts leading to occasional high emission rates.

In the manuscript, we will include how our rewetting scenarios relate to practical, real-life management practices.

2. Choosing one RCP is understandable considering the amount of variables that are already present in the study. I do not suggest that you introduce a milder climate scenario here but I think it would be good to acknowledge in the discussion that this is the scenario leading to strongest impacts of climate change. This is particularly relevant for your manuscript, as you assess the cascading impacts of two climate-related variables. It would be highly interesting to know (perhaps in a future paper) if this relationship of WTD and Ta cancelling each other out is visible in other climate change scenarios, or would the influence of one overpower the other.

We agree in your comment regarding the choice of the RCP leading to strongest impacts of climate change. We will not introduce additional climate scenarios in the manuscript, but we

will highlight in the discussion that this scenario corresponds to the strongest climate change impacts.

3. Throughout the paper, you refer to the ground surface as "terrain", i.e. on L63 and L129. Is there a particular reason why you don't just say "surface"? I'm not sure I've ever heard this use of the term terrain, and at least for me, this was somewhat confusing.

**We will change the term from terrain to surface.**

4. I would advise on renaming the two future time periods as "mid-century" (or something like that" and "end of century". Distant future, to me at least, seems to be something further in the future than within my own lifespan, and since your simulation covers this century, it seems like a much clearer choice to refer to that.

**We will rename the future time periods in line with your suggestion.**

Fig.3: Could you make the lines within the legend a little thicker, so that it is clear which line (blue or black) is referring to modelled and measured values?

**Certainly, we will.**

Finally, while the language is generally very good, I would advise a thorough read-through of the manuscript, or employing the help of a proof-reader. I have listed below some parts that need refining, but I did not conduct a thorough language check.

Thank you for your language corrections. We will proofread the manuscript carefully, possibly with support from a professional proofreader.

L70. Do not account for neither: double negative.

L85: Sentence starting with "For example through..." is not a full sentence.

L91: field-scale, not field scale

L113: Should this be "continuous", not "contiguous"?

L126: This probably should be "automatically", not "automatic"

L306: There is no need to say "such as rewetting", since this is the only management practice discussed in this paper.

L311-313: This is a confusing sentence.

L657: "In 2023,  $CO_2$  emissions from drained organic soils in croplands and grasslands was estimated to

accounted for 6.7% of Denmark's total emissions" --> "was estimated to have accounted for"

L687: "Such nature-based solutions are not likely to reduce..."

---

## Author Response (AR1)

Dear Associate Editor

We have prepared a revised manuscript that incorporates the reviewers' comments as well as the points addressed in our response to their feedback.

Best regards

Tanja Denager and co-authors

---

## Author Response (AR2)

Reply to editor:

The manuscript improved after the review, still there are many issues with language, abbreviation, and inconsistencies in units.
Please carefully revise the manuscript.

In response to reviewer 1 about the real-world application of the authors added the following sentence (lines 676-683):

"Nature-based approaches represent the most common real-world rewetting strategies, aiming to restore peatlands towards their natural hydrological regime. At a minimum, such rewetting requires terminating tillage activities and eliminating artificial drainage for instance by blocking of drainpipes and ditches. The rewetting scenarios implemented in this study, represented as simple modifications to WTD, are not reflective of practical management interventions - except perhaps in a few rare and costly restoration projects that involve installing artificial impermeable membranes along peatlands edges (Naturstyrelsen, 2022). However, the outcome of this study can serve as a reference for discussions on realistic expectations on CO2 emission reductions from rewetted peatlands."

Although the sentence clarifies some aspects, such as the fact that the tested rewetting solution is only partly applicable in the real world due to the high costs, I think the authors should explain more clearly how the study can serve as a reference. How can the study serve as a benchmark? How can other solutions be implemented, and what would the expected results be compared to the benchmark? Is the selected strategy the one that can achieve the greatest emission reduction potential? Can you compare it with other studies and provide more context on this aspect?

What we mean by serving as a reference is that this study illustrates that the WTD dynamics after rewetting are very important for the impact on $CO_2$ emission reductions. So irrespective of the exact rewetting practice, this study introduces the consideration to post-restoration WTD dynamics and its influence on emission reductions. If rewetting mainly impacts winter WTD and the farmer still requires e.g. the area to be dry for cattle grazing in summer, and the rewetting supports that, then the $CO_2$ emission reduction is much lower that if the area is wet during summer. Commonly, within rewetting in Denmark, there is a perception that, when abandoning intensive farming on a peatland area and seizing drainage, the natural hydrology will return it to a wet state where $CO_2$ emissions go to zero. Our hope is that our study can facilitate discussions of rewetting impacts and the need to monitor rewetting impacts on WTD dynamics and designing rewetting to increase WTD also during summer and drought periods where emissions are large.

We suggest reformulating the revised session above to illustrate this instead of merely state that it can "serve as a reference".

"However, the outcome of this study can inform discussions on requirements and best practices for rewetting and peatland restoration. The study also highlights the need to monitor or model pre- and post-restoration WTD dynamics in order to develop realistic expectations regarding $CO_2$ emission reductions from rewetted peatlands.

We are not aware of other studies in a Danish context that addresses the temporal aspect of $CO_2$ emission reductions, those are typically estimates by fixed emission factors for soil and land cover or by the annual WTD emission model described in the paper.

On line 57, there is something wrong with the parentheses.

Corrected

Line 88: 'most LSM's' should be 'most LSMs'.

Corrected

On line 170, 'LAI' please define LAI.

Corrected

I have noticed that units are sometimes reported as $kg/m^3$ or $kg\ m^{-3}$. Please use only the second throughout the manuscript.

Corrected.

Define KGE before line 202.

Between lines 202 and 230, the description is a mixture of equations and undefined terms. Please rewrite it, making sure that all the symbols and acronyms are defined, and that this part of the text is written as an equation (for example, the beta equation at line 207).

We agree lines 202-230 are a bit unorganized. However, the equations are all defined and described and written in equation form in Table 1. We suggest adding the definition of all variables in the equation inside Table 1 instead of writing all 5 equations in full inside the main text. These statistics are very common, they are KGE, ME and correlation coefficients, so we feel it is a bit too much to take up a large part of the text with five inserted equations and the associated definitions of variables. We prefer spending time explaining how and why we

combined the objective functions in the way we did for this calibration exercise. We have edited the section a bit to avoid the intext equation and introduced table 1 earlier which we believe will also guide the reader better.

Figure S1: Clarify what Kristensen and Jensen mean.

The reference to Kristensen and Jensen (1975) is included in the caption

The units are missing from Figure S3. Please ensure that you add them to the axes.

Now in caption

Line 263: Please clarify what MgCO2-C refers to. I believe it is cumulative, but it should be defined.

It is the annual $CO_2$ emission (Mg $CO_2$-C ha$^{-1}$ yr$^{-1}$). See figure 5.

Line 324: I suggest changing the title to 'Uncertainty of future climate $CO_2$ emissions'.

We agree, but prefer 'Uncertainty of future $CO_2$ emission estimates', to not confuse it with the uncertainty of the climate model global emission scenarios.

Line 451: This is the third time that the NECB has been introduced.

Corrected.

Figure 5: Please add the uncertainty of the fitting to the figure.
Expect for the green and yellow dots figure 5 is based solely on work by (Koch et al., 2023) and (Tiemeyer et al., 2020). We consider it outside the scope of this manuscript to evaluate the uncertainty of the models that originates from separate studies.

However, we have proceeded to calculate the uncertainty for this reply.

Below we show a version of figure 5 where the uncertainty is included. At the figure three different fits of the Annual WTD model are shown; (Elsgaard, 2024; Koch et al., 2023; Tiemeyer et al., 2020). In the manuscript we chose to use the (Koch et al., 2023) curve , as it appears in a peer-reviewed scientific journal (which (Elsgaard, 2024) doesn't) and is derived from Danish flux data. The figure below shows that the three Annual WTD models are very similar.

Uncertainty is estimated by performing 1000 bootstrap resamples (with replacement) of the Danish and German flux dataset (n=145). A Gompertz function is then fitted to each of the

1000 resampled flux datasets, and the uncertainty is expressed as ±1 std.dev of the (Elsgaard, 2024) curve. As mentioned above, it is important to note that the *Annual WTD model* used in this manuscript is based solely on the Danish flux data (Koch et al., 2023), which is why it is not centered within the blue band.

[Figure]

Line 709: RCP has already been defined above.

Corrected.

Reference:

Elsgaard, L.: Dokumentations notat vedr. forskningsprojekter om analyse af danske emissionsdata ( > 12 pct . OC ) samt relation mellem emission fra jorder med 6-12 pct . OC og > 12 pct . OC, Rådgivningsnotat fra DCA - Nationalt center for fødevarer og jordbrug, Århus University, 2024.

Koch, J., Elsgaard, L., Greve, M. H., Gyldenkærne, S., Hermansen, C., Levin, G., Wu, S., and Stisen, S.: Water-table-driven greenhouse gas emission estimates guide peatland restoration at national scale, Biogeosciences, 20, 2387–2403, https://doi.org/https://doi.org/10.5194/bg-20-2387-2023, 2023.

Kristensen, K. J. and Jensen, S. E.: A model for estimating actual evapotranspiration from potential evapotranspiration, Nordic Hydrology, 170–188, 1975.

Tiemeyer, B., Freibauer, A., Borraz, E. A., Augustin, J., Bechtold, M., Beetz, S., Beyer, C., Ebli, M., Eickenscheidt, T., Fiedler, S., Förster, C., Gensior, A., Giebels, M., Glatzel, S., Heinichen, J., Hoffmann, M., Höper, H., Jurasinski, G., Laggner, A., Leiber-Sauheitl, K., Peichl-Brak, M., and Drösler, M.: A new methodology for organic soils in national greenhouse gas inventories: Data synthesis, derivation and application, Ecol Indic, 109, 105838, https://doi.org/10.1016/j.ecolind.2019.105838, 2020.